# From Finite Element Analysis to Self-Optimization: FRAME (Finite element Reasoning and AI-Agent Model Engine) and the Era of Agentic AI in Engineering

## Mahish K. Guru[14], Vipul Gupta[12], Kartik Bali[13], Roland C. Aydin[13]

[1]Helmholtz-Zentrum Hereon, Geesthacht; [2]Brandenburg University of Technology; [3]Hamburg University of Technology; [4]Leuphana University Lüneburg

Finite Element Analysis (FEA) is a cornerstone of aerospace, automotive, civil, biomedical engineering, etc. However, the extensive visual and numerical outputs it generates—such as stress fields, strain distributions, and temperature maps—create a major bottleneck in the design cycle. Interpreting this data demands specialized expertise and deep contextual domain knowledge to make informed design decisions.[1] Multimodal large Language Models (LLMs) and Vision Language Model (VLMs), presents a transformative opportunity to automate this critical step. These models excel at complex reasoning and may interpret scientific visuals and physics effectively.[2] Yet, in computational mechanics, their progress is limited by missing domain-specific benchmarks. General datasets lack the nuanced semantics of engineering physics, causing a mismatch between model abilities and field needs. [3] [4]

This paper introduces FRAME (Finite element Reasoning and AI-Agent Model Engine) to address this gap. Our primary contributions are twofold: FRAME Benchmark and FRAME application in Self Optimization Agentic Workflow.

FRAME Benchmark: We present a new, open-source benchmark dataset designed specifically for the interpretation of FEA results. It pairs visual simulation outputs like contour plots and key problem parameters (e.g., material properties, boundary conditions)—to a structured textual description. This provides a robust foundation for evaluating and fine-tuning models on tasks of

- Identifying critical features like stress concentrations, yielding, or maximum deformation,

- Extracting critical values (e.g., maximum von Mises stress) and

- Proposing a specific, actionable change to the component's geometry or material to address an identified issue (e.g., "increase fillet radius to reduce stress concentration").

The dataset is composed of thousands of data points systematically curated from academic literature and in-house simulations, covering a diverse range of engineering

domains, including aerospace, automotive, and biomechanics. The quality of the dataset was validated through human expert evaluation on criteria such as faithfulness, physical accuracy, and completeness, confirming its suitability for both model evaluation and fine-tuning.

We evaluated the zero-shot performance of several leading general-purpose multimodal language models on the FRAME benchmark. We also fine-tuned a suite of smaller open-source models on the FRAME training set, resulting in slight improvement in the evaluation metrics. Using semantic similarity and ROBERTA-based keyword-based accuracy measures, these fine-tuned models outperformed the larger general-purpose baselines, underscoring the importance of domain-specific data.

**Self Optimization Agentic Workflow:** We contextualize these fine-tuned models within a proposed agentic workflow for automated design improvement, showcasing a path toward autonomous, self-optimizing engineering systems.[5][6]

The enhanced reasoning capabilities of models fine-tuned on FRAME lay the foundation for a broader vision: a closed-loop, agentic system for automated design optimization. In this paradigm, the fine-tuned vision-language model serves as the core reasoning engine of a multi-step workflow. The system begins by analyzing FEA results from an initial design iteration, identifying key weaknesses such as stress concentrations or excessive deformation. It then proposes specific, parameterized design modifications—for instance, increasing a fillet radius to reduce stress. Acting on these insights, another agent generates a Python script to update the constructive solid geometry (CSG) of the CAD model and triggers a re-simulation using the modified geometry and boundary conditions.

This iterative loop refines designs based on simulation feedback, shifting engineers from manual analysis to supervisory roles. The result is faster design space exploration and quicker convergence to optimal solutions. FRAME is essential for fine-tuning large AI models to meet the complex demands of engineering, advancing self-optimizing digital twins and reducing R&D time and cost.

However, limitations exist. The current scope of FRAME is primarily focused on Finite Element Analysis. Future work will expand the benchmark to include more complex physics, such as fluid dynamics, as multi-scale material simulation data. An immediate next step is the development of a Retrieval-Augmented Generation (RAG) agent that combines the reasoning power of the fine-tuned model with a vast knowledge base of design handbooks and material datasheets, further enhancing its decision-making capabilities.

[1] Patil, Onkar & Saxena, Shruti. (2025). Automation and Data Analysis in FEA Post-Processing Onkar ravindra Patil. International Journal of Emerging Technology and Advanced Engineering.

[2] Kononenko, Oleksiy & Kononenko, I.. (2018). Machine Learning and Finite Element Method for Physical Systems Modeling. https://doi.org/10.48550/arXiv.1801.07337.

[3] Lu, W., Luu, R.K. & Buehler, M.J. Fine-tuning large language models for domain adaptation: exploration of training strategies, scaling, model merging and synergistic capabilities. *npj Comput Mater* **11**, 84 (2025). https://doi.org/10.1038/s41524-025-01564-y.

[4] Mianroodi, J.R., H. Siboni, N. & Raabe, D. Teaching solid mechanics to artificial intelligence—a fast solver for heterogeneous materials. *npj Comput Mater* **7**, 99 (2021). https://doi.org/10.1038/s41524-021-00571-z.

[5] Timms, A., Langbridge, A., Antonopoulos, A., Mygiakis, A., Voulgari, E., & O'Donncha, F. (2025). Agentic AI for Digital Twin. *Proceedings of the AAAI Conference on Artificial Intelligence*, *39*(28), 29703-29705. https://doi.org/10.1609/aaai.v39i28.35373.

[6] Tim Kreuzer, Panagiotis Papapetrou, Jelena Zdravkovic,Artificial intelligence in digital twins—A systematic literature review,Data & Knowledge Engineering,Volume 151,2024,ISSN 0169-023X,https://doi.org/10.1016/j.datak.2024.102304.
