# OpenReview forum: "From Finite Element Analysis to Self-Optimization: FRAME (Finite element Reasoning and AI-Agent Model Engine) and the Era of Agentic AI in Engineering"
_NLDL.org/2026/Abstracts_Track — NLDL 2026 Abstracts_

### Official Review · Reviewer_hwFK · 2025-10-27

**Soundness:** 3
**Correctness:** 3
**Rating:** 4
**Confidence:** 4

**Summary:**

The abstract introduces FRAME (Finite element Reasoning and AI-Agent Model Engine), a benchmark and workflow framework designed to advance the use of multimodal large language models (LLMs) and vision-language models (VLMs) for interpreting finite element analysis (FEA) results. FRAME addresses the challenge of automating the complex, expert-driven post-processing stage in engineering by providing a domain-specific dataset and agentic workflow, which together allow AI models to suggest and execute design improvements autonomously. The work highlights both the limitations of current general-purpose models in this domain and the potential for specialized benchmarks and workflows to enable self-optimizing engineering systems.

**Strengths:**

Novelty & Significance: The creation of a domain-specific benchmark for FEA interpretation fills a clear gap, enabling more effective evaluation and fine-tuning of AI models in engineering contexts.

Technical Validity: The approach of pairing visual outputs and key parameters with structured textual descriptions provides a strong foundation for developing and testing AI capabilities in mechanical reasoning.

Comprehensive Scope: The dataset covers a wide range of engineering domains (e.g., aerospace, automotive, biomechanics), increasing its generalizability and value to the community.

Practical Impact: By embedding fine-tuned models in an agentic, closed-loop workflow for design optimization, this work has the potential to significantly improve design cycles, reduce manual workload and speed up convergence to optimal solutions.

Evaluation: The claims are supported by benchmarking against leading multimodal models, with evidence that domain-specific fine-tuning improves performance.

Vision for Extension: The authors acknowledge current limitations and outline concrete directions for expanding FRAME, e.g., into more complex physics and retrieval-augmented design.

**Weaknesses:**

Current Scope Limitation: The present benchmark is focused mainly on FEA and does not yet address other critical engineering simulations, such as fluid dynamics or coupled multi-physics, which may limit broader adoption.

Experimental Depth: While initial improvements from fine-tuning are demonstrated, the quantitative results are only briefly mentioned. A deeper analysis, including more examples or failure cases, would strengthen the evaluation.

Reproducibility: Details about the size, diversity, and specific curation process of the dataset are limited. More information about benchmark construction and access could help others reproduce/extend this work.

Agentic Automation: The abstract discusses the vision of a self-optimizing workflow, but evidence of real-world deployment or rigorous simulation-in-the-loop testing is limited.

Generality & Robustness: It remains to be seen how robust the proposed approach is to new geometries, load cases, or manufacturing constraints, and whether it could generalize beyond standard problems.

---

### Official Review · Reviewer_zEKu · 2025-10-27

**Soundness:** 3
**Correctness:** 4
**Rating:** 4
**Confidence:** 4

**Summary:**

The abstract presents a solution for practical tasks applicable across multiple research areas using Finite Element Reasoning and an AI-Agent Model Engine. The authors effectively combined domain knowledge to mitigate the impact of limited data and incorrect labeling in Finite Element Analysis.

**Strengths:**

The authors demonstrate strong skills in integrating Multimodal Large Language Models (MLLMs) and Vision Language Models (VLMs) to develop a Retrieval-Augmented Generation (RAG) system with self-optimizing capabilities. They explored related research fields to guide the application of their model and transparently discussed its limitations.

**Weaknesses:**

- Follow the NLDL conference template.

- For the poster, include more informative results such as tables and images to better illustrate findings.

- Specify the model parameters that were tuned.

- Strengthen the literature review.

- Report computational efficiency, including at least an approximate order of magnitude for saved computation time.

---

### Decision · Program_Chairs · 2025-11-05

**Decision:**

Accept

**Comment:**

The abstract is of interest to the community and should be presented at the conference.